Clinical validation of the short and long UNESP-Botucatu scales for feline pain assessment

Belli Maíra 1
de Oliveira Alice R. 2
de Lima Mayara T. 1
Trindade Pedro H.E. 2
Steagall Paulo V. 1 3
Luna Stelio P.L. stelio.pacca@unesp.br 2
1 Department of Surgical Specialties and Anesthesiology / Medical School, São Paulo State University (Unesp) , Botucatu , São Paulo , Brazil
2 Department of Veterinary Surgery and Animal Reproduction / School of Veterinary Medicine and Animal Science, São Paulo State University (Unesp) , Botucatu , São Paulo , Brazil
3 Département de Sciences Cliniques / Faculté de Médecine Vétérinaire, Université de Montréal , Saint-Hyacinthe , Québec , Canada
Aly Sharif
Electronic publication date: 2021 Apr 12
Publication date: 2021
Volume: 9
Electronic Location ID: e11225
Received 2020 Sep 28; Accepted 2021 Mar 16
Copyright: ©2021 Belli et al.
Copyright year: 2021
Copyright holder: Belli et al.
License: This is an open access article distributed under the terms of the Creative Commons Attribution License, which permits unrestricted use, distribution, reproduction and adaptation in any medium and for any purpose provided that it is properly attributed. For attribution, the original author(s), title, publication source (PeerJ) and either DOI or URL of the article must be cited.
License URL: https://creativecommons.org/licenses/by/4.0/

Keywords: Analgesia, Animal welfare, Feline, Orthopedics, Pain, Pain scale, Postoperative care, Reliability, Validation, Cat

Funding: FAPESP 2017/12815-0 This work was suported by Capes scholarship for Maíra Belli and São Paulo Research Foundation (FAPESP) thematic research project (2017/12815-0). The funders had no role in study design, data collection and analysis, decision to publish, or preparation of the manuscript.

==============================
Background

The UNESP-Botucatu multidimensional feline pain assessment scale (UFEPS) is a valid and reliable instrument for acute pain assessment in cats. However, its limitations are that responsiveness was not tested using a negative control group, it was validated only for ovariohysterectomy, and it can be time-consuming. We aimed to evaluate the construct and criterion validity, reliability, sensitivity, and specificity of the UFEPS and its novel short form (SF) in various clinical or painful surgical conditions.

Methods

Ten client-owned healthy controls (CG) and 40 client-owned cats requiring pain management for clinical or surgical care (20 clinical and 20 surgery group (12 orthopedic and eight soft tissue surgeries) were recruited. Three evaluators assessed pain, in real-time, in clinical cases before and 20 min after rescue analgesia and in surgical cases before and up to 6.5 hours postoperatively, by using the visual analog, numerical ratio, and a simple descriptive scale, in this order, followed by the UFEPS-SF, UFEPS and Glasgow multidimensional feline pain (Glasgow CMPS-Feline) in random order. For the surgical group, rescue analgesia (methadone 0.2 mg/kg IM or IV and/or dipyrone 12.5 mg/kg IV) was performed when the UFEPS-SF score was ≥4 or exceptionally according to clinical judgement. If a third interventional analgesia was required, methadone (0.1–0.2 mg/kg IM) and ketamine (1 mg/kg IM) were administered. For the clinical group, all cats received rescue analgesia (methadone 0.1–0.2 mg/kg IM or IV or nalbuphine 0.5 mg/kg IM or IV), according to the clinician in charge, regardless of pain scores. Construct (1—comparison of scores in cats undergoing pain vs pain-free control cats by unpaired Wilcoxon-test and 2—responsiveness to analgesia by paired Wilcoxon test) and concurrent criterion validity (Spearman correlation of the total score among scales), inter-rater reliability, specificity and sensitivity were calculated for each scale (α = 0.05).

Results

Reliability ranged between moderate and good for the UFEPS and UFEPS-SF (confidence intervals of intraclass coefficients = 0.73–0.86 and 0.63–0.82 respectively). The Spearman correlation between UFEPS and UFEPS-SF was 0.85, and their correlation with Glasgow CMPS-Feline was strong (0.79 and 0.78 respectively), confirming criterion validity. All scales showed construct validity or responsiveness (higher scores of cats with clinical and postoperative pain vs healthy controls, and the reduction in scores after rescue analgesia). The sensitivity and specificity of the UFEPS, UFEPS-SF and Glasgow CMPS-Feline were moderate (sensitivity 83.25, 78.60% and 74.28%; specificity 72.00, 84.67 and 70.00%, respectively).

Conclusions

Both UFEPS and UFEPS–SF showed appropriate concurrent validity, responsiveness, reliability, sensitivity, and specificity for feline acute pain assessment in cats with various clinical and orthopedic and soft tissue surgical conditions.

Introduction

Being free from pain, injury, and disease is one of the fundamental five freedoms in animal welfare (Mellor, 2016; Robertson, 2018). The International Association for the Study of Pain (IASP) defines pain as “an unpleasant sensory and emotional experience associated with, or resembling that associated with, actual or potential tissue damage”. In addition, the “inability to communicate does not negate the possibility that a human or a nonhuman animal experiences pain” (Raja et al., 2020).

Pain management in animals has improved over time due to significant advances in its recognition, assessment and treatment (Lorena et al., 2013; Lorena et al., 2014; Robertson, 2018). However, animals still do not receive adequate analgesic therapy due to the lack of validated pain assessment scoring systems. Indeed, many veterinarians feel insecure and unable to correctly identify the presence of pain in animals (Lorena et al., 2013; Lorena et al., 2014).

In cats, the development and validation of species-specific, composite, and multidimensional pain scales have influenced feline pain recognition and assessment. These instruments may also provide clinical decision aid for analgesic provision once reaches a certain threshold (i.e., rescue analgesia) (Brondani et al., 2012; Brondani et al., 2013a; Brondani et al., 2013b; Merola & Mills, 2016; Reid et al., 2017; Steagall & Monteiro, 2019; Evangelista et al., 2019). They often include physiological and behavioral parameters associated with the observational and dynamic interaction between the cat and the observer.

There are currently three scales with reported validation for evaluating pain in cats: the UNESP-Botucatu multidimensional pain assessment scale (UFEPS) (Brondani, Luna & Padovani, 2011; Brondani et al., 2012; Brondani et al., 2013a; Brondani et al., 2013b), the Glasgow Feline Composite Measure Pain Scale (Glasgow CMPS-Feline) (Calvo et al., 2014; Reid et al., 2017), and the Feline Grimace Scale (Evangelista et al., 2019; Evangelista et al., 2020; Watanabe et al., 2020).

The UFEPS is the only one with reported validation (i.e., construct, content and criterion validation, reliability and sensitivity) in several languages such as Portuguese (Brondani et al., 2012; Brondani et al., 2013a), English (Brondani, Luna & Padovani, 2011; Brondani et al., 2013b), Spanish (Brondani et al., 2014), French (Steagall et al., 2017), and Italian (Della Rocca et al., 2018). The instrument is available on the website http://www.animalpain.com.br in Portuguese, Spanish, and English for didactic and scientific training purposes. The UFEPS is divided into subscales such as pain expression, psychomotor changes, physiological variables, and miscellaneous behaviors. The use and application can be time-consuming and complex, and the scale was only validated for cats undergoing ovariohysterectomy (Brondani et al., 2013b). It is not known whether the tool could be applied to other types of surgical or clinical pain. Additionally, it requires blood pressure monitoring which is not always feasible and/or practical in the clinical setting. Recently, a short version of the UFEPS (UFEPS–SF) has been developed in eight different languages to overcome these limitations and facilitate clinical pain assessment in feline practice (Luna et al., 2020). The instrument has easy applicability and has been used in a previous clinical trial (Benito et al., 2019). The UFEPS-SF consists of four items (0–3 points for each item) to evaluate the cats’ posture, activity, attitude, and reaction to touch and palpation of a painful site. The items “appetite” and “blood pressure monitoring” were not included in the short form. The maximum total score is 12, and rescue analgesia is provided at ≥ 4. The instrument and video examples of behaviors of each item may be found at http://www.animalpain.com.br.

According to Merola & Mills (2016), the UFEPS is the “only specific instrument with evidence of validity, reliability and sensitivity at the level of a randomized control trial”. However, if one considers its aforementioned limitations and the lack of another gold-standard instrument for acute pain assessment in cats, the Glasgow CMPS-Feline was used for comparison in the current study because it had been gone through some degree of validation.

The objective of this study was to evaluate the validity (construct and criterion), reliability and sensitivity of the UFEPS and UFEPS-SF, compared with each other and with Glasgow CMPS-Feline for feline acute pain assessment in various painful clinical conditions, and after orthopedic and soft tissue surgery. The hypothesis of the study was that the UFEPS and UFEPS–SF are valid, reliable, and sensitive to the administration of analgesics in cats undergoing different painful clinical and surgical conditions.

Materials & Methods

This was a prospective, clinical, cohort study. It was carried out at the Veterinary Hospital of the School of Veterinary Medicine and Animal Science—São Paulo State University (UNESP)-Botucatu, Brazil, between March and December 2019. The study protocol was approved by the Ethics Committee on the Use of Animals of the same institution under protocol number 0039/2019. Written tutor consent for participation in the study was obtained for each cat.

Animals

A total of 53 mixed-breed male or female cats (Felis catus) of different age and body weight were enrolled for the study. Cats were enrolled into three groups: control pain-free cats (CG), surgery group (SG) and clinical nonsurgical group (ClinG). Ten client-owned clinically healthy cats, without any painful conditions were recruited for the control group (CG) from the Veterinary Hospital personnel. Forty-three client-owned cats requiring health care were admitted to the Veterinary Hospital and enrolled in the study after physical and often laboratory and imaging examination required according to the clinician’s decision. The inclusion criteria for the surgery group were cats needing a surgical procedure, and that could tolerate the anesthetic and surgical procedure. Exclusion criteria were cats with feral/aggressive behavior, cats that left the Veterinary Hospital before a total of six hours of observations and cats that required post-surgery intensive care. Forty cats met the inclusion criteria. Twenty cats with medical conditions (Clinical Group—ClinG) and 20 cats undergoing surgery [(Surgery Group—SG); divided into two subgroups if they underwent orthopedic surgery (OrthG; n = 12) or soft tissue surgery (SoftG; n = 8)] were included in the study (Fig. 1).

The CG and SG animals were housed individually in a stainless-steel cage, 120 cm wide, 60 cm high, and 60 cm deep with a litter box, bed, and blanket. Water and food were offered ad libitum, except during fasting before surgery (pre-operative of SG). A maximum of two cats was evaluated simultaneously. The animals in the ClinG were evaluated at the primary care service in the owner’s presence and, when possible, inside a stainless-steel cage 60 cm wide, 60 cm high, and 60 cm deep. Otherwise, the pain was assessed with the animal on the examination table.

Pain assessment and rescue analgesia

Three veterinarians evaluated the cats in real-time and in-person: the main observer (MB—completing a MSc. program) and two other graduate students (PhD and MSc) who had previously completed a residency program in Veterinary Anesthesiology (ARO and MTL). Evaluators were trained to use the UFEPS scale by assessing the http://www.animalpain.com.br website to observe the behaviors that corresponded to each item of the scale and by assessing their ability to use the scale. Evaluators did not have access to each one’s scores during evaluation. Initially, they observed the cats from a distance without interaction. Then, the main observer interacted with the animal and performed the physical examination, including pain assessment, and the others only observed. Pain assessment was performed with the unidimensional numerical ratio (NS; 0 “no pain” to 10 “worst possible pain”), simple descriptive (SDS; 1 “no pain” to 4 “worst possible pain”) and visual analog scales (VAS—horizontal line of 10 cm length where “0” corresponds to “no pain” and 10 cm to “worst possible pain”) in this order, followed by the composite scales UFEPS–SF (Table 1), UFEPS (Brondani et al., 2013b) and the Glasgow CMPS-Feline (Reid et al., 2017) in random order. The composite scales are based on several categories, including descriptive levels of behaviors graded from 0 to 3 (UFEPS and UFEPS-SF) or 0 to 4. The six possible orders for assessing the three composite scales were randomized (randomization.org) and excluded after use until all possible orders were used, followed by a new randomization process. The subscale “physiological variables” (i.e., appetite and blood pressure) of the UFEPS was not included in the assessment because repetitive blood pressure monitoring can be stressful in cats, especially in those with painful conditions. In relation to appetite, some cats at some time points could not eat as they were fasted preoperatively. According to the original UFEPS (Brondani et al., 2013b), the subscales may be assessed separately because there is an intervention analgesic point calculated for each subscale. A Portuguese version of the Glasgow CMPS-Feline was translated from English by the authors and used in the study (Reid et al., 2017) and then back-translated by a non-veterinarian independent translator to ensure semantic equivalence. The time to assess each composite scale was recorded in one cat from each group.

Figure 1 Flowchart of cats included in the study.

Red rectangle: animals included in the analysis of responsiveness to rescue analgesia.

Table 1 Short form of the UNESP-Botucatu multidimensional feline pain assessment scale (UFEPS-SF).

Item	Description	Score	
Evaluate the cat’s posture in the cage for 2 min.	
1	Natural, relaxed and/or moves normally	0	
Natural but tense, does not move or moves little or is reluctant to move	1	
Hunched position and/or dorso-lateral recumbency	2	
Frequently changes position or restless	3	
		Please tick where applicable	
2	The cat contracts and extends its pelvic limbs and/or contracts its abdominal muscles (flank)		
The cat’s eyes are partially closed (do not consider this item if present until 1 h after the end of anesthesia)		
The cat licks and/or bites the affected area		
The cat moves its tail strongly		
All above behaviors are absent	0	
Presence of one of the above behaviors	1	
Presence of two of the above behaviors	2	
Presence of three or all of the above behaviors	3	
Evaluation of comfort, activity and attitude after the cage is open and how attentive the cat is to the observer and/or surroundings	
3	Comfortable and attentive	0	
Quiet and slightly attentive	1	
Quiet and not attentive. The cat may face the back of the cage	2	
Uncomfortable, restless and slightly attentive or not attentive. The cat may face the back of the cage	3	
Evaluation of the cat’s reaction when touching, followed by pressuring around the painful site	
4	Does not react	0	
Does not react when the painful site is touched, but does react when it is gently pressed	1	
Reacts when the painful site is touched and when pressed	2	
Does not allow touch or palpation	3	

Surgery group

The SG cats were admitted 30 to 60 min before the procedure and discharged up to seven hours after surgery. Premedication was performed with methadone (Mytedom®, Cristália, Itapira, São Paulo, Brazil) or methadone and xylazine (Anasedan®, Ceva, Paulínia, São Paulo, Brazil), anesthesia was induced with propofol (Propovan®, Cristália, Itapira, São Paulo, Brazil) alone or combined with ketamine (Dopalen®, Ceva, Paulínia, São Paulo, Brazil) and/or fentanyl (Fentanest®, Cristália, Itapira, São Paulo, Brazil) and maintained with isoflurane alone (Isoforine®, Cristália, Itapira, São Paulo, Brazil) or combined with intravenous (IV) ketamine and/or fentanyl (Table S1). Cats were evaluated immediately before premedication (baseline) and at every hour from 1 to 6 h (or 6.5 h in case cats received dipyrone for rescue analgesia; see below) after extubation (Fig. 2A). The analgesic intervention was performed when the UFEPS-SF score was ≥ 4 out of a total score of 12 points (Benito et al., 2019) using methadone (0.2 mg/kg intramuscularly; IM or IV, if an intravenous catheter was available) and/or dipyrone (Analgex V®, Agener União, São Paulo, Brazil - 12.5 mg/kg; IV) both diluted up to a volume of 1 mL. The choice between the two drugs was based on the observers’ clinical decision and pain intensity. In exceptional cases, when the observers felt that the cats could be in pain, analgesia was provided even if the UFEPS-SF score was <4/12. Pain was assessed 60 and 90 min after administration of methadone and dipyrone respectively, according to the drug pharmacokinetics (Slingsby et al., 2016; Lebkowska-Wieruszewska et al., 2018). If required, the second administration of rescue analgesia consisted of methadone for cats that had received dipyrone and vice-versa. If a third interventional analgesia was required, a combination of methadone (0.1–0.2 mg/kg IM) and ketamine (1 mg/kg IM) was administered. Rescue analgesia was not provided if cats presented signs of dysphoria (restlessness, vocalization, and agitation) within the first hour after extubation. If cats were not painful at 2 or 4 h after surgery, pain assessment at consecutive moments (3 and 5 h) was not performed to minimize the stress of handling.

Figure 2 Timeline of the study, time-points for pain assessment and rescue analgesia.

Pain was evaluated with NS, numeric; SDS, simple descriptive; VAS, visual analog; UFEPS, UNESP-Botucatu multidimensional feline pain assessment scale, UFEPS-SF, short version of the UNESP-Botucatu scale, and Glasgow CMPS-Feline, Glasgow feline multidimensional pain scale. (A) Surgery Group. Time points for pain assessment varied according to the drug used for rescue analgesia. (B) Clinical Group. All cats received rescue analgesia. (C) Control Group. *In the case of signs of dysphoria, rescue analgesia was not performed. **Evaluation was not performed in cases when cats were painless and comfortable at the previous time point. 1 Reevaluation after 60 min. 2 Reevaluation after 90 min. 3 If procedural sedation was required for further diagnostics, pain scores after the administration of sedatives were not included in the analysis of construct validity.

Clinical group

Twenty cats suffering pain from trauma produced by fracture (n = 7) or soft tissue damage (n = 3), abdominal pain due to lower urinary tract disease (n = 4) or fecaloma (n = 1), abdominal and sacral pain due to the trauma (n = 1), recent penectomy (n = 1), osteosynthesis (n = 1), migration of a pin after ulna osteosynthesis (n = 1) and abscess in the left hind limb (n = 1) (Table S1) were evaluated immediately before and 20 min after administration of rescue analgesia in all cases (Fig. 2B). Rescue analgesia with methadone (0.1–0.2 mg/kg IM or IV) or nalbuphine (Nubain®, Cristália, Itapira, São Paulo, Brazil −0.5 mg/kg IM or IV) was selected according to the clinician’s decision. Cats were excluded if procedural sedation was required for further diagnostics. In this case, pain scores after the administration of sedatives were not included in the construct validity analysis (responsiveness to rescue analgesia).

Control group

The CG cats were admitted 30 min before the first assessment and discharged after the last assessment. The evaluations were performed at 0 (30 min after admission), 30, 60, 90, and 120 min (Fig. 2C).

Statistical analysis

Statistical analyses were performed using R software in the RStudio integrated development environment (RStudioTeam, 2016) (Table 2). For all analyses, an α of 0.05 was observed for statistical significance. A minimum sample size of 10 was calculated based on a difference of 3 points of the total score (standard deviation = 3) in UFEPS-SF before and after rescue analgesia (http://biomath.info/power/). The description of all tests is in Table 2. The Shapiro Wilk test confirmed that data did not have a normal distribution, therefore nonparametric tests were used for all analysis. The inter-rater reliability was calculated by intraclass correlation coefficients (ICC) to detect the agreement of the total score of each scale among the observers (Bartko, 1966). Comparisons were made in pairs of two observers at each time using a one-way model that considered each cat as a random effect . The concurrent criterion validity was calculated by Spearman’s coefficient to estimate the correlation between the total scores of all scales. Responsiveness was calculated by two ways: unpaired Wilcoxon test was used to test the hypothesis that animals in the surgery and clinical groups had higher pain scores than in the control group and second the paired Wilcoxon test was used to test the hypothesis that scores after analgesia were lower than those before analgesia. Sensitivity and specificity were estimated to detect if the scales could detect the true positives corresponding to cats with pain (surgical and clinical groups) and true negatives corresponding to cats without pain (control group) (see Table 2 for a detailed description of statistical analysis).

Table 2 Statistical analyses used for validation of the short (UFEPS-SF) and long (UFEPS) versions of the UNESP-Botucatu multidimensional feline pain assessment scales.

Analysis Type	Description	Database	Test	
Inter-rater reliability	A matrix was generated to assess the agreement of the total score of each scale among the observers.	All observers (3), cats (n = 50), groups and time points were used (7 time points in Surgery group—before surgery, 1, 2, 3, 4, 5 and 6 h after recovery from anesthesia; 2 time points in Clinical group—before and 20 min after rescue analgesia; and 5 time points in the Control group—0 (30 min after admission), 30, 60, 90, and 120 min.	For the UFEPS, UFEPS-SF and Glasgow CMPS-Feline, intraclass coefficient (ICC) ”consistency” type was used and its 95% confidence interval (CI). For the NS and SDS the weighted kappa coefficient was used. The 95% CI kw (“cohen.kappa” function of the “psych” package) was estimated. For the VAS, intraclass coefficient (ICC) “agreement” type was used and its 95% CI (”icc” function of the ”irr” package). Interpretation of values: <0.5 poor; 0.5–0.75 moderate; 0.75–0.9 good; >0.9 excellent (Koo & Li, 2016).	
Concurrent criterion validity	Correlation of the total score between all scales at all time points.		Spearman rank correlation coefficient (r; “rcorr” function of the “Hmisc” package). Interpretation of the degree of correlation: <0.19 very weak; 0.2–0.39 weak; 0.4–0.59 moderate; 0.6–0.79 strong; 0.8–1—very strong (Evans, 1996).	
Construct validity (responsiveness to control group)	The responsiveness of each scale was determined by testing the hypothesis that animals in the SG and ClinG groups have higher pain scores than in the CG.	The time point with the highest UFEPS score for each cat before rescue analgesia was selected from the main evaluator. The database included the scores of all evaluators at these same highest UFEPS score time points [3 evaluators × 40 time points/cats (20 cats from SG; OrthG, n = 12 and SoftG, n = 8 and 20 cats from ClinG) = 120)] were compared to the scores of all evaluators assessed at 120 min in CG (3 evaluators x 1 time point × 10 cats = 60).	Analyses were performed for the OrthG, SoftG, and ClinG separately, as well as for the SG and ClinG together. Unpaired Wilcoxon test was used to compare scores (“wilcox.test” function of the “stats” package).	
Construct validity (responsiveness to rescue analgesia)	Responsiveness and effect of time (sensitivity to change) was determined for all scales by testing the hypothesis that scores after analgesia are lower than those before analgesia.	Only one-time point before and one-time point after rescue analgesia of cats that received analgesia were selected from SG (SG, surgery; OrthG, orthopedic, n = 11 and SoftG, soft tissue surgeries, n = 5), and those that did not receive sedation from ClinG were included (clinical, n = 13) (3 evaluators × 29-time points = 87).	Analyses were performed for the OrthG, SoftG, and ClinG separately, as well as for the SG and ClinG together. The paired Wilcoxon test was used to compare the scores before and after analgesia (“wilcox.test” function of the “stats” package).	
Sensitivity of the scale	Based on true positives—cats with pain (surgical and clinical groups).	272 and 244 time points of grouped SG and ClinG when UFEPS scores were ≥ 7/24 and Glasgow CMPS-Feline were ≥ 5 for the three evaluators were used as database respectivelly. From these time points, the number of time points at which each scale had their score ≥ the cut-off point was filtered (cats that were supposedly feeling pain—true positives) and divided by these time points	Sensitivity = True positives/Total number of time points (“ci.coords” function of the “ROCR” package).
Interpretation: excellent 100–95%; good 94.9–85%; moderate 84.9–70%; not sensitive <70% (Streiner & Norman, 2008).	
Specificity of the scale	Based on true negatives—cats without pain from the control group (CG).	All time points of all evaluators of CG (5 time points × 10 cats × 3 evaluators = 150). The calculation was based on the number of time points when each scale presented a score lower than the cut-off point (cats that were supposedly not feeling pain - true negatives) divided by the total number of time points.	Specificity = True negatives/Total number of time points (“ci.coords” function of the “ROCR” package).
Interpretation: excellent 100–95%; good 94.9–85%; moderate 84.9–70%; not specific <70% (Streiner & Norman, 2008).	
Notes.

Groups: SG, surgery group (OrthG, orthopedic; SoftG, soft tissue); ClinG, clinical; CG, control group. Cut-off points for calculation of sensitivity and specificity: numeric (NS) ≥ 4/10, simple descriptive (SDS) ≥ 2/4, visual analog (VAS) > 28/100 (Brondani et al., 2013a), the short version of the UFEPS (UFEPS-SF) ≥ 4/12, UNESP- Botucatu multidimensional feline pain assessment scale (UFEPS) ≥ 7/24 and Glasgow feline multidimensional pain scale (Glasgow CMPS-Feline) ≥ 5/20 (Reid et al., 2017).

Results

Forty-three client-owned, mixed-breed cats with clinical conditions or undergoing surgery were enrolled in the study (Fig. 1, Table S1). One feral/aggressive behavior and two cats that left the Veterinary Hospital before the six-hour observational period postoperatively were excluded. Another ten client-owned, mixed-breed, clinically healthy cats without any painful conditions were included for the CG (Fig. 1).

Of all 50 animals recruited, 17 were female (34%) and 33 males (66%), aged 3.8 ± 4.3 years with a bodyweight of 3.8 ± 1.3 kg. Analgesic, anesthetic, surgical procedures, and clinical conditions are described in Table S1.

Duration of pain assessment using the UFEPS-SF, Glasgow CMPS-Feline, and UFEPS (excluding appetite and blood pressure) was 90, 112 and 166 s, respectively. In the OrthG, 11 of 12 cats required the administration of rescue analgesia. In the SoftG, 5 out of 8 cats required rescue analgesia. Seven from 20 cats of the ClinG required procedural sedation and chemical restraint; their pain scores were not included after the administration of sedatives (Fig. 1). Sixteen cats required analgesia in the SG for 31 times (29 times using the criteria of UFEPS–SF scores ≥ 4 and 2 by using clinical judgment).

Inter-rater reliability

The inter-rater reliability of the unidimensional scales (NS, SDS, and VAS) was moderate (confidence interval values 0.57–0.78). For the composite scales, reliability ranged from moderate to good (0.63–0.86) (Table 3).

Concurrent criterion validity

The correlations between unidimensional (NS, SDS, and VAS) versus Glasgow CMPS–Feline, UFEPS, and UFEPS-SF were weak to moderate (0.48–0.52), moderate (0.54–0.58) and moderate to strong (0.58–0.62), respectively. The correlations were strong between the Glasgow CMPS-Feline and both UFEPS scales (0.78–0.79). The correlations were very strong between the unidimensional scales (0.88–0.93) and between the UFEPS and UFEPS-SF (0.85) (Table 4).

Construct validity (responsiveness to the control group)

The scores of the surgery and clinical groups together or alone (OrthG, SoftG and ClinG) were significantly higher compared with the controls (Table 5), which characterizes the responsiveness of all scales compared to controls.

Construct validity (responsiveness to rescue analgesia)

For all groups, the scores after the administration of analgesia were lower than the ones before (Table 6). For this analysis, four cats from the SG (one from the OrthG and three from the SoftG) were excluded because they did not need rescue analgesia in the postoperative moment, and seven cats from the ClinG were excluded because they required procedural sedation or chemical restraint (Fig. 1).

Table 3 Inter-rater reliability matrix for the total scores of unidimensional and multidimensional scales to assess pain in cats.

Scales	Observer 1vs:	Observer 2vsObserver 3	
	Observer 2	Observer 3		
	Kappa (confidence interval)	
NS	0.74 (0.74–0.74)	0.67 (0.67–0.67)	0.73 (0.73–0.73)	
SDS	0.66 (0.66–0.66)	0.6 (0.6–0.6)	0.6 (0.6–0.6)	
	Intraclass correlation coefficient type agreement (confidence interval)	
VAS	0.65 (0.57–0.73)	0.72 (0.64–0.78)	0.7 (0.62–0.76)	
	Intraclass correlation coefficient type consistency (confidence interval)	
UFEPS-SF	0.77 (0.71–0.82)	0.76 (0.69–0.81)	0.71 (0.63–0.77)	
UFEPS	0.81 (0.76–0.85)	0.79 (0.73–0.84)	0.82 (0.77–0.86)	
Glasgow CMPS-Feline	0.72 (0.65–0.78)	0.74 (0.67–0.79)	0.77 (0.71–0.82)	
Notes.

Database: All observers (3), cats (n = 50), groups and time points were used (7 time points in Surgery group—before surgery, and 1, 2, 3, 4, 5 and 6 hours after recovery from anesthesia; 2 time points in Clinical group—before and 20 minutes after rescue analgesia; and 5 time points in the Control group—0 (30 minutes after admission), 30, 60, 90, and 120 minutes. Scales: NS, numeric; SDS, simple descriptive; VAS, visual analog; UFEPS-SF, short version of UFEPS; UFEPS, UNESP-Botucatu multidimensional feline pain assessment scale; Glasgow CMPS-Feline, Glasgow feline multidimensional pain scale. Interpretation of values: <0.5: poor; 0.5–0.75: moderate; 0.75–0.9: good; >0.9: excellent (Koo & Li, 2016). (p = 0.000001).

Sensitivity of the scales

Considering all time points (total of 485) and including observations from the three evaluators, there were 272 and 244 time points at which the total score of UFEPS and Glasgow CMPS-Feline were equal or above their cut-off point (≥ 7/24 and ≥ 5/20, respectively) for the administration of analgesics in the SG and ClinG. The sensitivity was as follows for the other scales respectively (meaning that rescue analgesia would have also been administered if the cut-off for these instruments had been used at these same time points): good for SDS (90.04% and 88.83%) and moderate for UFEPS-SF (79.66% and 78.60%), VAS (78.84% and 79.54%), and NS (76.77% and 75.34%). Sensitivity was 74.28% for Glasgow CMPS-Feline and 83.25% for UFEPS when they were compared to each other (Table 7).

Table 4 Spearman’s correlation matrix between the total scores of unidimensional and multidimensional scales to assess pain in cats.

	NS	SDS	VAS	UFEPS-SF	UFEPS	
SDS	0.92					
VAS	0.93	0.88				
UFEPS-SF	0.61	0.58	0.62			
UFEPS	0.58	0.54	0.58	0.85		
Glasgow CMPS-Feline	0.49	0.48	0.52	0.78	0.79	
Notes.

Database: All observers (3), cats (n = 50), groups and time points were used (7 time points in Surgery group—before surgery, and 1, 2, 3, 4, 5 and 6 hours after recovery from anesthesia; 2 time points in Clinical group—before and 20 minutes after rescue analgesia; and 5 time points in the Control group—0 (30 minutes after admission), 30, 60, 90, and 120 minutes. Scales: NS, numeric; SDS, simple descriptive; VAS, visual analog; UFEPS-SF, short version of the UNESP-Botucatu scale; UFEPS, UNESP-Botucatu multidimensional feline pain assessment scale; Glasgow CMPS-Feline, Glasgow feline multidimensional pain scale. Interpretation of the degree of correlation: <0.19: very weak; 0.2–0.39: weak; 0.4–0.59: moderate; 0.6–0.79: strong, 0.8–1: very strong (in bold) (Evans, 1996).

Table 5 Median and range pain scores of unidimensional and multidimensional scales for feline pain assessment of the control cats and of cats with clinical or postoperative pain.

Scales	CG	OrthG	SoftG	ClinG	SG + ClinG	
	Median (range)	Median (range)	pvalue	Median (range)	pvalue	Median (range)	pvalue	Median (range)	pvalue	
UFEPS	2.5 (0–17)	9 (3–14)*	<0.01	9 (1–17)*	<0.01	10 (2–18)*	<0.01	9 (1–18)*	<0.01	
UFEPS-SF	1 (0–6)	5 (1–9)*	<0.01	5 (1–9)*	<0.01	5 (2–9)*	<0.01	5 (1–9)*	<0.01	
Glasgow CMPS-Feline	2 (0–10)	7 (1–13)*	<0.01	7 (1–15)*	<0.01	7.5 (2–14)*	<0.01	7 (1–15)*	<0.01	
VAS	1.5 (0–10)	51 (12–96)*	<0.01	66 (1–97)*	<0.01	56 (15–98)*	<0.01	56 (1–98)*	<0.01	
SDS	1	3 (1–4)*	<0.01	2.5 (1–4)*	<0.01	3 (1–4)*	<0.01	3 (1–4)*	<0.01	
NS	1	5.5 (1–10)*	<0.01	6 (1–10)*	<0.01	6 (3–10)*	<0.01	6 (1–10)*	<0.01	
Notes.

Time point with the highest UFEPS score for each cat before rescue analgesia was selected from the main evaluator. Database included the scores of all evaluators at these same highest UFEPS score time points [3 evaluators × 40 time points/cats (20 cats from SG—OrthG, n = 12 and SoftG, n = 8—and 20 cats from ClinG) = 120)] were compared to the scores of all evaluators assessed at 120 min in CG (3 evaluators × 1 time point × 10 cats = 60). Groups: SG, orthopedic and soft tissues surgeries; OrthG, orthopedic surgeries; SoftG, soft tissue surgeries; ClinG, clinical. Scales: NS, numeric; SDS, simple descriptive; VAS, visual analog; UFEPS-SF, short version of the UFEPS; UFEPS, UNESP-Botucatu multidimensional feline pain assessment scale; Glasgow CMPS-Feline, Glasgow feline multidimensional pain scale.

* Significant difference compared with CG according to the Mann-Whitney test (p < 0.05).

Specificity of the scales

Of the 150 time points evaluated in the CG of all the observers grouped, the unidimensional scales demonstrated 100% of the scores below the cut-off point and, therefore, excellent specificity. The specificity of the UFEPS–SF, UFEPS and Glasgow CMPS-Feline was moderate (84.67%, 72.00%, and 70.00% respectivelly) (Table 7).

Table 6 Median and range scores of unidimensional and multidimensional scales for feline pain assessment before and after the administration of analgesics.

Scales	OrthG	SoftG	ClinG	SG + ClinG	
	Before	After	pvalue	Before	After	pvalue	Before	After	pvalue	Before	After	pvalue	
UFEPS	10 (3–14)	6 (0–10)*	<0.01	10 (4–17)	6 (2–14)*	<0.01	10 (2–18)	7 (1-14)*	<0.01	10 (2–18)	6 (0–14)*	<0.01	
UFEPS-SF	5 (1–9)	2 (0–9)*	<0.01	6 (2–9)	3 (1–8)*	<0.01	5 (2–8)	3 (0–8)*	<0.01	5 (1–9)	3 (0–9)*	<0.01	
Glasgow CMPS-Feline	7 (2–11)	3 (0–7)*	<0.01	9 (3–15)	4 (0–6)*	<0.01	7 (2–14)	4 (0–9)*	<0.01	7 (2–15)	4 (0–9)*	<0.01	
VAS	48 (12–96)	28 (7–65)*	<0.01	71 (2–97)	46 (2–58)*	<0.01	56 (15–95)	26 (2–78)*	<0.01	55 (2–97)	27 (1–78)*	<0.01	
SDS	3 (1–4)	2 (1–3)*	<0.01	3 (1–4)	2 (1–3)*	<0.01	3 (1–4)	2 (1–3)*	<0.01	3 (1–4)	2 (1–3)*	<0.01	
NS	5 (1–10)	3 (1–6)*	<0.01	7 (1–10)	4 (1–7)*	<0.01	6 (1–9)	3 (1–7)*	<0.01	6 (1–10)	3 (1–7)*	<0.01	
Notes.

Database: Only one-time point before and one-time point after rescue analgesia of cats that received analgesia were selected from SG (SG, surgery; OrthG, orthopedic, n = 11 and SoftG, soft tissue surgeries, n = 5), and those that did not receive sedation from ClinG were included (clinical, n = 13) (3 evaluators × 29-time points = 87). Scales: NS, numeric; SDS, simple descriptive; VAS, visual analog; UFEPS-SF, short version of the UFEPS; UFEPS, UNESP-Botucatu multidimensional feline pain assessment scale; Glasgow CMPS-Feline, Glasgow feline multidimensional pain scale

* Significant difference compared to before analgesia according to Wilcoxon test (p < 0.05).

Table 7 Sensitivity and specificity of the unidimensional and multidimensional scales to assess pain in cats.

	Sensitivity (%)	Specificity (%)	
	Based in UFEPS	Based in Glasgow CMPS-Feline				
Scales	Estimated	CI (95%)	Estimated	CI(95%)	Estimated	CI(95%)	
		Min.	Max.		Min	Max		Min	Max	
NS	76.77	70.91	81.94	75.34	69.03	80.95	100.00	100.00	100.00	
SDS	90.04	85.54	93.51	88.83	83.84	92.72	100.00	100.00	100.00	
VAS	78.84	73.13	83.82	79.54	73.52	84.71	100.00	100.00	100.00	
UFEPS-SF	79.66	74.02	84.57	78.60	72.51	83.88	84.67	77.89	90.02	
UFEPS				83.25	77.58	87.98	72.00	64.09	79.02	
Glasgow CMPS-Feline	74.28	68.27	79.68				70.00	61.99	77.20	
Notes.

Database—sensitivity: 272 and 244 time points of grouped SG and ClinG when UFEPS scores were ≥ 7/24 and Glasgow CMPS-Feline were ≥ 5, respectively. The number of time points at which each scale had their score the cut-off point was filtered (cats that were supposedly feeling pain—true positives) and divided by 272 and 244 moments. Specificity: all time points of all evaluators of CG (5 time points × 10 cats × 3 evaluators = 150). The calculation was based on the number of moments when each scale presented a score lower than the cut-off point (cats that were supposedly not feeling pain—true negatives) divided by the total number of time points. Scales: NS, numeric; SDS, simple descriptive; VAS, visual analog; UFEPSSF, short version of the UNESP-Botucatu scale; UFEPS, UNESP-Botucatu multidimensional feline pain assessment scale; Glasgow CMPS-Feline, Glasgow feline multidimensional pain scale. Interpretation: excellent 100–95%; good 94.9–85%; moderate 84.9–70%; not sensitive or specific <70% (Streiner & Norman, 2008).

Discussion

This study showed that the UFEPS and UFEPS–SF are valid, reliable, responsive, sensitive, and specific scoring systems. The study fills a gap by highlighting the reliability and responsiveness of the UFEPS for a pain assessment in cats with different clinical and postoperative pain conditions, including orthopedic surgeries. This overcomes the previous limitation that the scale had been developed and validated only for postoperative pain associated with ovariohysterectomy. In addition, a group of healthy, control cats were included to corroborate the responsiveness of the scale confirmed by finding that pain scores were significantly higher in painful versus pain-free cats. Similar findings were also obtained for the UFEPS-SF. Therefore, this simplified, user-friendly version of the UFEPS can be readily used in feline practice and overcomes another limitation of the UFEPS: the scale is not cumbersome and time-consuming. It can also be used in eight languages (Luna et al., 2020).

The inter-rater reliability of the UFEPS was predominantly good across observers (ICC of 0.79–0.82); however, it was lower when compared to the original study in cats undergoing ovariohysterectomy (ICC of 0.98 for the total score and 0.93 to 0.97 for subscales 1 and 2, interpreted as excellent) (Brondani et al., 2013a; Brondani et al., 2013b). In another study, the inter-rater reliability of the UFEPS for observers with different degrees of experience was moderate (0.7) with great variability in CI (0.2–0.89) (Benito et al., 2017), suggesting that training may affect the reliability of these pain scoring systems. On the other hand, the inter-rater reliability of the Glasgow CMPS-Feline and UFEPS-SF was moderate to good (0.65–0.82 and 0.63–0.82, respectively). We hypothesize that these scales have less detailed descriptors and worse reliability when compared with the UFEPS. However, this should not have a major clinical impact; these instruments can be used in different pain conditions if they are used by individuals with experience in pain assessment.

Inter-rater reliability was moderate for the unidimensional scales (NS, SDS, and VAS). Similar ICC results were found when a dynamic and interactive VAS was used by observers with different experiences (Benito et al., 2017). However, their CI values were lower (0.19 to 0.8) compared to the current study (0.57 to 0.78). Unidimensional pain scales may not capture all the complexity of pain (Robertson, 2018). Nevertheless, they might be acceptable for clinical pain assessment when used by experienced observers.

The correlation of a new scale with instruments with reported validation (i.e., gold-standard) is required in the study of concurrent criterion validity (Streiner & Norman, 2008). The correlation of the UFEPS–SF with the UFEPS and Glasgow CMPS-Feline was very strong and strong, respectively, and confirms the criterion validity for this scale. In previous studies, criterion validity was very strong between the Feline Grimace Scale and the Glasgow CMPS-Feline (0.86) without the facial component (Evangelista et al., 2019) and strong between the UFEPS and the Glasgow CMPS–Feline (0.6–0.8) (Steagall et al., 2018). The criterion validity between the UFEPS and UFEPS-SF with the Feline Grimace Scale should be a subject of a future study.

The ability to detect a significant change in pain scores, whether by decreasing the score after rescue analgesia or increasing the score after a painful procedure, is part of construct validity or responsiveness of the instrument (Von Baeyer & Spagrud, 2007; Chien et al., 2013). It determines whether the scale can detect differences between known groups (e.g., painful versus non-painful individuals) (McDowell, 2009; Brondani et al., 2013b). Such comparisons have already been used in the validation of the Feline Grimace Scale and the Glasgow CMPS-Feline (Reid et al., 2017; Evangelista et al., 2019). The construct validity of the UFEPS was previously determined by comparing baseline scores with the highest pain postoperative scores (Brondani et al., 2013b). The absence of a negative control group in Brondani et al. (2013b) study was a limitation described in a systematic review (Merola & Mills, 2016). All scales used in the current study, including the UFEPS, distinguished animals with clinical and surgical pain from those without pain, demonstrating the responsiveness of the UFEPS and UFEPS-SF for both the administration of rescue analgesia and in comparison, with controls. The limitation of the construct validity in the current study was that the observers were biased to the painful status of these cats. Indeed, they knew if cats had had surgery or required analgesic administration for pain relief. The same occurred for the control cats; the observers knew that these cats were most likely not suffering from any painful condition. Therefore, the scores given for each scale could be influenced by the assessments carried out previously. These limitations may have inflated the scores given before rescue analgesia and deflated the scores given after administering analgesia independently of the scale used. Both clinical real-time and image assessment of the Feline Grimace Scale (Evangelista et al., 2020) and video scoring with the UFEPS (Brondani et al., 2012; Brondani et al., 2013b) was reported in previous studies. Otherwise, construct validity of the UFEPS–SF should be further corroborated in a future study via video assessment with observers who are blinded to the analgesic administration and painful status.

Regarding sensitivity, the UFEPS–SF and Glasgow CMPS-Feline scales detected the most truly painful cases. Overall, the sensitivity of the unidimensional and multidimensional scales was moderate, showing that close to 80% of the cats suffering pain would be correctly diagnosed (true pain). Specificity showed that the three multidimensional scales detected most of the true negatives; the UFEPS-SF had the best specificity to identify pain-free cats. Although the Glasgow CMPS-Feline had similar results to the other multidimensional scales, it presented lower sensitivity and specificity than the other scales. One possible reason was the translation into Portuguese; however, back translation of the instrument ensured semantic equivalence. Translation and back-translation of a scale are required to validate a scale to be used in a different language. It is important for the semantics and terminology of the new instrument (Streiner & Norman, 2008; Sousa & Rojjanasrirat, 2011). Likewise, for responsiveness, the results for specificity were also biased in this study since observers were most likely aware of the cats’ painful status. Therefore, it is not surprising that the unidimensional scales had an excellent specificity (i.e., detection of true negatives; non-painful client-owned cats from the Veterinary Hospital personnel).

A limitation was that the results for the unidimensional scales might have been influenced by not randomizing them before the assessment and not performing their evaluations individually. Another limitation was the order of pain assessment; the scores of the unidimensional scales might have influenced the scores given in the subsequent composite scales. The authors decided to prioritize the assessment of the composite scales because this was the primary objective of the study. Pain assessment using the composite scales could have inflated the scores of the unidimensional scales had they been used first because they indicate which pain behaviors should be assessed (Roughan & Flecknell, 2006). Considering that unidimensional scales are subjective because they do not include pain behaviors in their assessments, the authors considered that unidimensional instruments would have less influence in subsequent composite scale assessments than the other way round. This approach was similar to other previously published papers in cats and other species (Brondani et al., 2013b; De Oliveira et al., 2014; Silva et al., 2020).

A possible confounding factor that may influence postoperative pain assessment is the use of sedatives and anesthetics. Indeed, pain could not be assessed in some cats one hour after the end of surgery due to the presence of residual anesthesia. Pain could be overestimated with false-positive results, as described anecdotally in dogs (Mathews et al., 2014). Under these circumstances, cats would receive unnecessary analgesia. The same applies to postoperative dysphoria and excitement. Most of the cats received ketamine for induction of anesthesia and some for intraoperative pain management. Ketamine increases psychomotor scores using the UFEPS, falsely increasing pain scores (Buisman et al., 2016). Another limitation of the study is that only animals presenting feral or aggressive behavior were excluded. Shy individuals were included. Shy and aggressive cats may have higher scores on the psychomotor subscale of UFEPS and Glasgow CMPS-Feline scales (without the facial component) (Buisman et al., 2017) due to their unique demeanor.

It is not known if pain scores were affected by the presence of observers and potential cat owners in this study. In a previous study (Evangelista et al., 2020) for the Feline Grimace Scale, there was no significant difference between real-time and video assessments. Real-time scores were slightly overestimated when compared with video scores, which would probably not affect the clinical assessment. These are some limitations that demonstrate the challenges of clinical pain assessment and the development and validation of pain scoring instruments in veterinary medicine.

In summary, possibly because unidimensional scales have no descriptors, they have less inter-rater reliability than the composite ones. However, concurrent criterium validity, responsiveness, sensitivity, and specificity were comparable for both unidimensional and composite scales. The UFEPS-SF had the lowest possibility to have pain-free cats diagnosed as painful and provided the quickest assessment time when compared to others. It is important to highlight that training was performed beforehand and that the robustness of theses scales should be assessed using untrained observers.

Conclusions

Both UFEPS and UFEPS–SF showed appropriate concurrent validity, reliability, responsiveness, sensitivity, and specificity for feline acute pain assessment in cats with various clinical conditions and those undergoing orthopedic and soft tissue surgery. The results of this study for the UFEPS-SF should be corroborated in a future study by using a masked and randomized design.

Supplemental Information

Data S1 Dataset used for all the statistical analysis

All pain scores from different pain assessment tools performed by the observers of all cats included in this study. These scores were used to validate the long and short UNESP-Botucatu scales for feline acute pain assessment.

Click here for additional data file.

Table S1 Analgesic, anesthetic, and surgical procedures of surgical and clinical groups

Click here for additional data file.

Supplemental Information 3 CONSORT flow diagram adapted

Click here for additional data file.

Supplemental Information 4 Adapted CONSORT checklist

Click here for additional data file.

Additional Information and Declarations

Competing Interests

Author Contributions

Animal Ethics

Data Availability

The authors declare there are no competing interests.

Maíra Belli conceived and designed the experiments, performed the experiments, analyzed the data, prepared figures and/or tables, authored or reviewed drafts of the paper, and approved the final draft.

Alice R. de Oliveira and Mayara T. de Lima performed the experiments, authored or reviewed drafts of the paper, and approved the final draft.

Pedro H.E. Trindade analyzed the data, authored or reviewed drafts of the paper, and approved the final draft.

Paulo V. Steagall conceived and designed the experiments, authored or reviewed drafts of the paper, and approved the final draft.

Stelio P.L. Luna conceived and designed the experiments, analyzed the data, authored or reviewed drafts of the paper, and approved the final draft.

The following information was supplied relating to ethical approvals (i.e., approving body and any reference numbers):

Ethics Committee on the Use of Animals (CEUA) of School of Veterinary Medicine and Animal Science (FMVZ) –São Paulo State University (Unesp)-Botucatu, Brazil approved this research (protocol number 0039/2019).

The following information was supplied regarding data availability:

The raw data measurements are available in the Supplementary File.

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
