# Peer review of "Clinical validation of the short and long UNESP-Botucatu scales for feline pain assessment"

_PeerJ, doi:10.7717/peerj.11225_

## Round 0.1 · original submission · Major Revisions

Thank you for submitting your research on feline acute pain assessment. In agreement with experts in the field its my opinion that the manuscript lacks clarity in methods. A good start would be to describe your study design, observational or experimental (intervention), also details on the frequency of observation varying by case type and pain assessment score are needed. Please consider clearly describing your methods and perhaps depicting the replicates (for case type, observers, or measures within patients) in a figure format along with the text, if possible. More clarity is also required when describing the random process as it is currently its confusing. Please address the reviewers comments line by line in your rebuttal identifying their requested information, statistical methods and data.

·

Basic reporting

Dear team, I would like to commend you on this research. Firstly, it's quite an ambitious comparison of multiple behavioural assessment scales, and must have taken a lot of thought to plan and execute. Secondly, the manuscript flows, is clear and easy to read.

One minor comment is that in the results section, for 'concurrent criterion validity' it would be good to define 'moderate', 'strong' and 'very strong' as you have done in other sections of the results.

Experimental design

1. I was slightly confused on the behavioural observation procedure, could you clarify against the following please:
As I read it, there were three observers, all in the room with the cat. one observer physically handled and manipulated the cat, while the other two looked on.
Is this correct?
Why then the reference to video recording in 147? Was this used at all in the current study, or is it stored for a future evaluation, as suggested in line 299.

2. You do correctly identify the challenges of bias in observers and the difficulty in blinding these observers when they are in the same room as the cat at the time of examination.
Another challenge is that the scores given in each 'test' could be influenced by the assessments made in the previous 'tests' carried out by they observer. You have randomised the order of UFAPS, UFAPS-SF and CMPS-Feline, but they all follow the unidimensional assessments (not randomised) - how would we correct for the potential for the unidirectional assessment scores to influence the scores given in the subsequent assessments?

3. It would be interesting to list the clinical conditions suffered by the ClinG cats.

Validity of the findings

No comment

Additional comments

Just a couple of very minor text edit suggestions:
Line 250, add 'pain' after 'postoperative'
Line 290, do you mean 'that study' rather than 'the UFAPS'?
line 291, suggest 'the current study' in place of 'this study'
Line 304, do you mean 'painful' instead of 'positive'?

·

Basic reporting

The current study titled “Clinical validation of the short and long UNESP-Botucatu scales for feline acute pain assessment (UFAPS) (#53086) is observational and comparing different pain assessment scales for cats with different medical conditions. The study adding more information about the validation of currently used feline pain assessment scales with the inclusion of a control group and cats with different surgical and medical cases. The authors evaluated the validity and reliability of the UFAPS and UFAPS-SF for feline acute pain assessment in comparison to the other four pain scales that have been used in cats. However, there are some issues that need to be revised:
1. It is not clear how the authors assessed the effect of time on all scales used. The repeated measure design was used in this study because the same animals were assessed over different times (moments). Each group was evaluated at different time points. For example, the clinical group evaluated at 2-time points, control at 5-time points, and the surgical group at 6-time points. However, the authors conducted concurrent validity based on data obtained from all time points. The difference between groups should be interpreted at each time point to obtain accurate validation.
2. The study was not blinded, and observers were aware of treatment groups which could have affected the obtained results. Also, the study was not randomized when unidimensional scales were used. Authors acknowledge that in the discussion section; however, authors should be cautious in their conclusion.
3. Different drugs with different pharmacological properties were used for anesthesia which could be a confounder that influences the obtained data of pain assessment. Authors should acknowledge that in their limitations.
4. It is not mentioned how the sample size was calculated, and what were inclusion and exclusion criteria in the study.

Specific comments:
Title: Using abbreviations in the title is not recommended. Please fix it.
Introduction:
Is well written and has enough information about the studied topic however the objective of the study needs rephrasing to make it more clear. What was the gold standard or reference used?
L89: Please provide a reference
L95: Please provide a website for this form as you did for the previous one
L99-100: Validated in comparison to what? Please clarify
Materials and Methods:
MMs need to be more organized with a logical flow of ideas. The rewriting of MMs could make it easier to read
L101: What do you mean by medical pain. Please explain
L111: Was approval/consent obtained from the owners of acts? You have it in L122, please move it to the first paragraph of MMs
L114: Please provide more information on how the sample size calculated? Did the authors perform power analysis before the start of the study?
L123-124: Please provide inclusion and exclusion criteria used in the study
L138: Please provide a reference and a brief description of each scale used in the study
L139: Change to CMPS-Feline to "Glasgow CMPS-Feline"
L142: Please explain why physiological variables were excluded from UFAPS during your evaluation?
L147: Which data was used for validation, the real-time assessment or video recorded data? Please clarify
L153: Throughout the whole manuscript whenever you mention drug please provide the source of drugs used in the study
L157: What do you mean by “the observers' discretion” please clarify
L159: Please state how many cases received analgesia with score of less than 4
L166: Please explain how the maintenance of anesthesia was achieved?
L168: Was the data from these animals included in statistical analysis?
L173: What do you mean by “clinician's discretion” please clarify
L180: What do you mean by last one. Please fix this sentence. What zero refers to in the control group?
Statistical analyses
Detailed information should be provided about the packages and functions used in R software. The authors should specify which tests were used in each software (R and Prism). Why did the authors use two different software for statistical analysis? Was the power analysis performed before the study? Was data checked for normality before analysis? It is not mentioned in table 2 which tests were used to calculate sensitivity and specificity.
The authors conducted concurrent criterion validity using all time points. However, each scale was evaluated at different time points. This is should be fixed in data analysis.
It will be helpful for R users and others to see a script of R commands and functions used in all different analyses, so results could be derived from the raw data provided in supplementary materials.

Results:
L188: It would be good to have a descriptive table for enrolled animals
L229&L239: Please provide a table for estimated sensitivity and specificity with (posterior median and 95% CI) of the conducted analysis
Tables: Please provide the sample sizes in the title of each table or as footnotes below tables as you have multiple cases of cat exclusion from data analysis
Discussion:
L259: You stated that “the scale is not cumbersome and time-consuming”. There was no data reported about the time required for conducting each scale. The authors should consider providing the time required for completing each assessment to be able to compare the reliability of each scale
L252: You stated that “pain scores were significantly higher in painful versus pain-free cats” Where is this data? The pain score data of each group was not presented in the results. Authors should include a table or figure showing the range of pain score in each group
L334: This the first time that author mentions the difference between real-time and video assessment. Where is this data? And how did you evaluate the difference? Please clarify
In the discussion section, authors should present the pros and cons of each scale, and based on their obtained results they can recommend one or more scales for clinics

Experimental design

There is a lot of limitations in the methodology that should be addressed.

Validity of the findings

Results are based on statistical analyses. However, some analyses can be done in a better way especially the effect of time.

Additional comments

The study is good work and obviously required plenty of effort and coordination for data collection. The verbiage needs to be more concise and better organized in some areas, especially in materials and methods.

---

## Round 0.2 · Minor Revisions

Thank you for addressing the reviewer comments. There are still some outstanding edits and details needed, please respond to the comments below or provide rebuttal.

Major comments:
Lines 223-224: The statistical methods are not provided still. Its fine to state that non parametric tests were used but you should detail them here, that Spearman's correlation coefficient (which is only mentioned in Table 4, otherwise not anywhere else in the text) was used to quantify the correlation between ...
Similarly detail the model or method used to estimate the ICC and provide the reference (so the R package, and the method statistical article, should be in the R vignette). If you actually ran a model, what was that model (given that the data is not normally distributed) and what was the random effect structure used.

Line 269: As you did in line 280 include the denominator.

Tables 5 and 6: Please replace all P values with <0.001 or better yet <0.01.

Table 7: I totally understand reporting the Se and Sp on the integer scale but not all lay readers will get it, either explain in the table title, or switch to percentage points. 2 decimal points are ideally needed (you currently dont report any decimal points since you are on the integer scale). Finally, are the CI's 95%?
Line 305: Please edit so you are not using slightly, its very subjective. Consider presenting the actual difference.


Minor comments:
Line 123: edit so it reads "Cats were enrolled into three groups.."
Line 137: last word should be were not was
Line 147: There is an extra period at start of the sentence
Line 194: "If a third.."
Line 220: edit so it reads: "... of 0.05 was observed for statistical significance"
Line 297: Edit so it reads: "predominantly good across observers." I see you followed this statement with the least lower confidence interval limit and the highest upper confidence interval, correct me if that is not the case but this is a bit strange but if you want to do that then you should explain what these are, otherwise its misleading to the readers as it will imply these are the 95% CI for a single estimate.

Line 349: drop the word cases, since all your subjects were cats (and cases cats sounds strange)

Line 374: replace confounder by confounding

·

Basic reporting

No comment - the manuscript has been satisfactorily revised

Experimental design

No comment - the manuscript has been satisfactorily revised

Validity of the findings

No comment - the manuscript has been satisfactorily revised

Additional comments

Thank you so much for your revisions, and your clear presentation of your approach to revisions.

·

Basic reporting

The paper is quite improved, and the authors have addressed all comments. The authors provided figure 2 that has detailed information about the methodology.
The manuscript looks good to me and I think it is ready for publication. Just minor changes to be addressed:
The authors abbreviated some words and did not use the abbreviations throughout the paper. Please check the paper for any abbreviations that are not used.
There are some grammatical errors, and extra spaces throughout the paper, so I am advising the authors to proofread the paper before final submission. Some references in the text are italic and others are not. Please fix throughout the paper.
L144 in word file, please change “any” with “different”.
L188 in word file, please delete the extra period.
L299-300 in word file: please provide the unit of time spent for assessment of each scale that was included in the results section.
L568 in word file, please change “patients” with “cats”.

Experimental design

NA

Validity of the findings

NA

Additional comments

See above.

---

## Round 0.3 · Minor Revisions

Thank you for addressing my comments, there is one pending edit that should be made and that is in lines 221-222:

"Comparisons were made in pairs of two observers at each time using a one-way model that considered each line as a random effect"

Technically speaking the line (data row) cannot be a random effect. Instead, this should describe the factor that is being assigned as a random effect. Is this the repeated measure over time in each cat or the observer? Please edit this so we can move with the acceptance.

---

## Round 0.4 · accepted · Accept

Congratulations! I find your manuscript acceptable for publication. Best wishes.